# Design and Modeling of a Terahertz Transceiver for Intra- and Inter-Chip Communications in Wireless Network-on-Chip Architectures

**DOI:** 10.3390/s24103220

**Published:** 2024-05-18

**Authors:** Biswash Paudel, Xue Jun Li, Boon-Chong Seet

**Affiliations:** Department of Electrical and Electronic Engineering, Auckland University of Technology, Auckland 1010, New Zealand; boon-chong.seet@aut.ac.nz

**Keywords:** system-on-chip, on-chip antenna, intra- and inter-chip communication, wireless network on chips, terahertz technology

## Abstract

This paper addresses the increasing demand for computing power and the challenges associated with adding more core units to a computer processor. It explores the utilization of System-on-Chip (SoC) technology, which integrates Terahertz (THz) wave communication capabilities for intra- and inter-chip communication, using the concept of Wireless Network-on-Chips (WNoCs). Various types of network topologies are discussed, along with the disadvantages of wired networks. We explore the idea of applying wireless connections among cores and across the chip. Additionally, we describe the WNoC architecture, the flip-chip package, and the THz antenna. Electromagnetic fields are analyzed using a full-wave simulation software, Ansys High Frequency Structure Simulator (HFSS). The simulation is conducted with dipole and zigzag antennas communicating within the chip at resonant frequencies of 446 GHz and 462.5 GHz, with transmission coefficients of around −28 dB and −33 to −41 dB, respectively. Transmission coefficient characterization, path loss analysis, a study of electric field distribution, and a basic link budget for transmission are provided. Furthermore, the feasibility of calculated transmission power is validated in cases of high insertion loss, ensuring that the achieved energy expenditure is less than 1 pJ/bit. Finally, employing a similar setup, we study intra-chip communication using the same antennas. Simulation results indicate that the zigzag antenna exhibits a higher electric field magnitude compared with the dipole antenna across the simulated chip structure. We conclude that transmission occurs through reflection from the ground plane of a printed circuit board (PCB), as evidenced by the electric field distribution.

## 1. Introduction

The ever-increasing requirements of computational power demands the addition of more processing units to a computer system. Processor design companies such as Intel and AMD are already utilizing System-on-Chips (SoCs), which allows for the incorporation of several cores within a single central processing unit (CPU). Intel Corporation announced a 14th-generation desktop processor in 2023, featuring 24 cores on a single chip [1]. The potential of a network-on-chip (NoC) allows for the integration of many cores into a single chip. Planar metal interconnect links with multi-hop packet switching communication are used for networking inside a chip. Metal wire (bus-based) connections (C4) are used to link cores within a chip (intra-chip) and across chips (inter-chip) [2]. The technology advancement in metal wire connection no longer satisfies the performance requirement for the communication inside a chip that is vital for high-power computing applications. Memory synchronization via metal wire connections may face issues in the near future when more cores are integrated into a chip. This increase in wire connections may increase the complexity of integration, leading to a higher manufacturing cost. Furthermore, as the number of cores in a device increases for HPC workloads, communication between distant cores requires many hops, where data packets will transit via a router/switch. This causes extra delay and power consumption, as well as a high signal-to-noise ratio (SNR) requirement [3].

A Wireless Network-on-Chip (WNoC) is a new concept that replaces conventional interconnects with wireless ones. By embedding an antenna into the chip of a multi-core system, communication may occur throughout the entire chip via a single hop, which can be either point-to-point or point-to-many, as in a broadcast system such as sharing cache memory [4]. WNoCs are made up of transmitters and receivers. Artificial intelligence (AI), virtual reality (VR), and augmented reality (AR) applications necessitate high-speed data processing, putting substantial strain on the chip’s processing units. Wireless networks linking these cores must support orders-of-magnitude-greater data speeds, in the hundreds of gigabits per second (Gbps) range. Terahertz (THz) technology appears as a viable alternative to satisfy the bandwidth and low-latency needs of these wireless links. THz antennas are particularly well suited for WNoC research because of their small size. A micron-sized wireless transceiver improves communications within the chip, enhancing the computational capability of multi-core processors [5].

In this study, we make a significant contribution to advancing communication within multi-core processor architectures by focusing on the implementation of wireless interconnects. Through rigorous modeling and simulation, we investigate the capabilities of THz antennas for ultra-short-distance communication both within and among chips. Specifically, we explore the potential of THz technology within the context of a realistic flip-chip package and analyze the results of electromagnetic wave propagation such as transmission parameter, path loss, and electric field distribution within the chips. Our contribution incentivizes CPU manufacturing companies to explore the feasibility of intra- and inter-chip communication via THz wireless links.

The paper is organized as follows: Section 2 provides the significance of the study, Section 3 reviews the related work, Section 4 discusses the importance of EM wave characteristics in chips, Section 5 describes the proposed WNoC design, Section 6 presents and discusses the simulation results, and finally, the paper is concluded in Section 7.

## 2. Significance of the Study

In the wired connection of a multi-core CPU, communication among the cores is achieved through metallic wires, forming the interconnected network topology. Common topologies include ring and mesh topologies [6], as depicted in Figure 1. Depending on the application and the need to improve communication performance among cores, various topologies are utilized, such as ring circulant [7], Spidergon STNoC [8], star, binary tree, butterfly, and torus [9].

The network faces challenges with high latencies as the network diameter increases. In any network with an N × N mesh configuration, the network diameter is typically 2N−1. Therefore, to establish communication between cores situated at the edge in a 3 × 3 configuration, wired connections would require four hops, resulting in relatively higher latency. Efforts have been made by [10] to address this, including proposing the flattened butterfly, which connects concentrated cores/nodes in all dimensions using dedicated links to reduce the network diameter where cores are separated by two hops [11]. However, connecting distant cores requires long wires, which suffer from signal attenuation in high-speed communication.

Numerous methods have been proposed to enhance network efficiency, such as inserting long-range links in existing mesh networks (also known as diametrical 2D-mesh interconnection [12]), implementing long-range wired links following a small-world (SW) graph [13], virtualizing channels to connect distant cores directly to avoid intermediate nodes [14], and introducing global lines (G-lines) for control signals [15]. Intermediate repeaters/switches/routers are necessary to reach indirectly connected or distant cores. Additionally, the limited bandwidth of metal wires makes them unsuitable for contemporary high-speed computing applications.

New solutions have been proposed to enhance core communication efficiency, such as through-silicon vias (TSVs) [16], which enable the 3D integration of multiple cores stacked atop one another in a single chip, connected through TSVs. However, this approach leads to higher heat densities in specific areas, resulting in cooling issues. These issues could be mitigated through the use of micro-cooling channels [17], but due to their taller and wider nature, they increase the processing power and data transfer delay when using TSVs [18].

Recently, new designs incorporating photonics and optical interconnections have emerged [19,20], enabling communication at the speed of light with reduced interference, low loss, and increased bandwidth. Despite the larger bandwidth offered by optical communications, optical modulators, even in wired optical fiber links, still operate at relatively slow speeds [16]. Additionally, auxiliary circuits such as modulators, demodulators, and electrical-to-optical converters introduce fabrication costs and pose unique challenges in design and packaging. Furthermore, scattering and diffracting phenomena are unavoidable in optical systems, and implementing broadcasting features proves to be challenging. Alternatively, Field Programmable Gate Arrays (FPGAs) offer adaptability, allowing the configuration of hops for specific functions. This adaptability reduces the number of hops among cores, enhancing performance and decreasing latency [21].

In a wired connection, the signal travels at the speed of light through metallic wires. However, the rate can be influenced by several factors, such as the material, thickness, width, and spacing of the wires, which contribute to the RC delay [22]. Similarly, in wireless interconnections like Wireless Network-on-Chip (WNoC), the delay is affected by factors such as signal attenuation, path loss, interference, and a higher likelihood of errors. Despite these challenges, the larger bandwidth available in the THz band enables higher data rates and supports parallel transmissions without physical interference. This is possible due to the small size of THz antennas, which can be densely packed into a small area [23]. Moreover, the capability for omnidirectional broadcasting in wireless setups reduces the time needed for memory synchronization and lowers power consumption compared with wired connections. It also decreases the processing time at intermediate nodes by significantly reducing the number of required hops. A study comparing wired and wireless interconnects in a system with 64 cores found that wireless interconnections improve bandwidth per core and energy efficiency per packet by 15% and 39%, respectively [24]. Additionally, using various multiplexing techniques can further increase the data rate.

## 3. Related Works

### 3.1. THz Radio Wave Generation and Advancements in WNoC

Over the last decade, various experimental setups have been conducted in the THz band involving the generation, transmission, and reception of THz signals. One such experiment, as described by [25], involved generating a frequency band between 0.585 and 0.653 THz, which was upconverted from 12.2 and 13.6 GHz using a 1 mW source. This source consisted of four cascaded frequency doublers followed by a frequency tripler, all based on biased Schottky diodes, feeding into a horn antenna. Likewise, another fixed wireless link, as reported by [26], uses a 0.24 THz carrier frequency for transmitting over a distance of 850 m. This setup integrated a commercial 20 GHz signal generator with a frequency multiplier. In [27], the authors focused on THz communication channel modeling and propagation. This involved the use of an autonomous transmitter and receiver that utilized a subharmonic mixer to upconvert a 10 GHz signal to 300 GHz. The signal was then transmitted using a horn antenna. Another experiment [28] achieved the transmission of an 8K resolution video using a resonant tunneling diode (RTD) paired with a bow-tie antenna at a 0.3 THz frequency. Meanwhile, the TeraNova testbed, developed by [29], demonstrated wireless communication at a 1.02 THz frequency using a PSG Keysight signal generator. The signal was upconverted to the THz band through a Schottky-diode-based frequency multiplier chain and transmitted and received using directional diagonal horn antennas. Lastly, a plasmonic nano-transceiver for the THz band proposed by [30] is based on a high electron mobility transistor (HEMT) and enhanced with graphene material.

Research conducted by [31] utilizes multiple antennas on a single substrate carrier to analyze intra-chip wireless communication within the 10–75 GHz frequency range. Another proposed system for a wireless communication system in the THz band by [32] employs OOK modulation and standard WR-3.4 horn antennas for transmitting and receiving in chip-to-chip communication systems. A novel wireless interconnect structure, WCube, proposed in [33], incorporates a single transmitting antenna and multiple receivers within a polyimide layer. This configuration demonstrates an improvement in a channel loss of 20–30 dB compared with silicon substrates and was analyzed using Ansys HFSS software in the 0.1–0.5 THz frequency bands. Additionally, chip-scale THz propagation modeling performed by [34] uses a Hertzian dipole antenna to achieve results of 0.95 Tbps with enhanced reliability, suggesting that thinning the underfill layer could improve the path gain. Lastly, a ray-tracing multipath propagation channel model developed by [35] for WNoC within a flip-chip package indicates that intra-chip channel capacity can exceed 1 Tbps over a transmission distance of 5 mm, which then decreases to 0.382 Tbps as the distance extends to 40 mm.

### 3.2. Security in Wireless Interconnect

The exploration of communication security is crucial due to the significant potential of Terahertz electromagnetic (EM) waves leaking into space through intra- and inter-chip channels. This not only impacts efficiency but also poses security risks such as eavesdropping (ED) or external interference disrupting signals. To address these concerns, Ref. [24] proposes a burst-error correction code to detect jamming attacks in wireless interconnects and suggests alternative routes. Additionally, in the case of eavesdropping (ED), a lightweight, low-latency data scrambling method is proposed. Another approach, known as Built-In Self-Test (BIST) as reported in [36], monitors wireless interconnects and can accurately detect jamming attacks with 99.87% accuracy while causing less than 3% bandwidth degradation. In [37], the authors introduce a novel detection and defense mechanism for WNoC against Denial of Service (DoS) attacks and Hardware Trojans (HT). This involves reconfigurable MAC and communication protocols, along with deploying an adversarial machine learning approach to detect attacks in the wireless network, achieving a detection accuracy of 99.87%.

## 4. Discussion

Achieving a robust wireless network within a chip-package design and implementation requires a through understanding of the principles driving the signal propagation inside the chip in addition to complex routing algorithms. Through the clarification of electric field distribution, path loss, and exploration of different THz antenna configurations, we hope to meet the requirements of energy-efficient and high-speed communications within a complex chip architecture.

## 5. Proposed WNoC Design

### 5.1. Design Concept

The basic idea of WNoC is to incorporate an on-chip antenna into the core. This antenna modulates a signal, which is subsequently transferred at the speed of light, avoiding the need for several chip hops. Another on-chip antenna demodulates the signal at the receiving end. Various proposed WNoC systems operate in the 30–300 GHz frequency band, with data rates of several Gbps.

To achieve greater bandwidth utilization, this system frequently incorporates modulation algorithms such as higher-order modulation, Quadratic Phase Shift Keying (QPSK), and Quadratic Amplitude Modulation (QAM). However, these schemes consume more power and require intricate transceiver circuits. On the contrary, in lower or non-coherent modulation techniques, such as on–off keying (OOK), the requirement for signal phase knowledge is not required and the receiver determines whether the transmitted symbol is a 0 or a 1 by comparing the power of the received signal strength against a threshold. Additionally, it can function with simpler transceiver circuits as depicted in Figure 2. It features a power amplifier (PA) and upconversion mixer on the transmitter side. On the receiver side, there is a low-noise amplifier (LNA), downconversion mixer, and injection-lock voltage-controlled oscillator (VCO) shared with the transmitter. This design eliminates the need for a power-hungry phase lock loop (PLL) by employing direct conversion and injection lock technology [38].

However, they come with limited bandwidth. To address these challenges, communication in the THz band, with its higher frequency bandwidth, becomes feasible [39]. The micrometer-sized THz on-chip antenna offers the advantage of high bandwidth and data rates in the terabit per second (Tbps) range.

Several existing designs relying on Radio Frequency Complementary Metal–Oxide–Semiconductor (RF-CMOS) technology are currently accessible. In contrast with wired transmission lines, wireless communication is achievable in an omnidirectional fashion, enabling single-hop communication through broadcasting. This not only significantly diminishes power consumption but also facilitates faster communications. WNoC employs various multiplexing methods such as Frequency Division Multiplexing (FDM), Time Division Multiplexing (TDM), and Spatial Multiplexing (SDM) to enhance the data rate of communications. These techniques contribute to the efficiency and speed of wireless communication within the network-on-chip architecture. To meet the low-latency requirements of a WNoC, additional logic functionalities should be implemented to determine whether signals should propagate through wired connections or wireless ones.

In order to share the antenna among many cores, a routing algorithm proposed by [40], called the radio access control mechanism (RACM), reduces communication delay by 30% and saves energy by 25% under real and synthetic traffic scenarios. Another algorithm proposed by [41] utilizes a hybrid form of communication, i.e., wired and wireless interconnection, depending on the distances between the cores, thereby improving communication efficiency. A routing technique called adaptive WNoC, developed by [42], aims to reduce traffic at wireless routers by monitoring buffer occupancy. The results show significant improvement in packet latency as the network size increases. In the event of wireless interconnect failure, which poses significant risk and increases delay, a novel adaptive fault-tolerant wireless routing algorithm proposed by [43] routes packets through alternative hubs with high average hop counts, demonstrating improvements in congestion and resilience. A new routing algorithm called HoenyWiN, proposed in [44], outperforms traditional mesh-type networks in terms of delay, throughput, and energy consumption. This routing algorithm avoids long multi-hop wired paths by utilizing the nearest wireless interconnect.

In this paper, we employ a chip that has nine cores, as illustrated in Figure 3, with each core having an embedded antenna integrated into it so that each core could have its own wireless interconnect. For instance, wired connections between Core A and Core B are ideal for communication since they are located in the proximity. Conversely, wireless communication between Core A and Core I is preferable, as it allows for a single hop [45].

### 5.2. Antenna

The THz band is notorious for its high path loss, which poses challenges. To address this, an array of antennas appears promising, but it requires additional feeding circuits. Research has demonstrated that carbon-based materials like graphene or carbon nanotube (CNT) antennas support the propagation of surface plasmon polaritons (SPPs) across a wide range, from microwave to the infrared region. These materials offer advantages in terms of tunability, bandwidth, and size. However, with current fabrication methods, where chips are integrated using the CMOS process, using graphene material presents additional challenges. Various antenna structures, such as triangular monopoles [31], dipoles, patches [46], and zigzags [2], have been previously studied at lower frequencies of up to 70 GHz. Some have also been explored at mm-Wave and THz frequencies, but there is a lack of full chip-scale modeling and analysis of results, such as E-field distribution, path loss, and budget link examples, in high-frequency communication within chip cores and between chips. In this paper, a dipole and monopole zigzag antenna has been chosen due to its small size and omnidirectional radiation pattern, leading to easy integration in the chip. The specifications for these dipole antennas are illustrated in Figure 4, and their $|S_{11}|$ parameter in the THz band is shown in Figure 5. The on-chip zigzag antenna is 353 µm long, while the dipole antenna is 149.9 µm long, for operation in the THz band. For simulation purposes, a lumped port is applied to excite the antenna, and the size of the antenna is optimized using Ansys HFSS to resonate in the THz band. The $|S_{11}|$ parameter of both antennas under a polyimide substrate indicates that the dipole antenna has a wider bandwidth compared with the zigzag antenna, while both antennas exhibit approximately the same value of $|S_{11}|$.

### 5.3. Antenna Configuration inside a Chip

In this paper, we consider a chip with a typical dimension of 20 × 20 mm^2^ consisting of various layers of materials available in a standard chip. The basic layers of a chip are shown in Figure 6, and its properties in Table 1. At the top of the chip, heat sinks and heat spreaders play a crucial role in dissipating heat away from the chip due to their excellent thermal conductivity. Materials such as silicon carbide (SiC), beryllium oxide (BeO), and aluminum nitride (AlN) are commonly used as heat spreaders. The silicon layer serves as the foundation for the transistors, while the polyimide layer facilitates electromagnetic propagation due to its low dielectric constant. A more detailed structure can be found in [46].

In a multi-core processor, interconnect wires are typically made out of copper and are arranged orthogonally to each other, following a Manhattan architecture [47], and they can be categorized into three main types: global, semi-global, and local [2]. Global interconnects are primarily used for tasks such as power supply, clock distribution, or long-range communications, covering substantial distance within the processor. Semi-global interconnects serve inter-block communication within a single processing core, typically covering shorter distances compared with global ones. Finally, local interconnects are responsible for connecting transistors within a unit, serving a more localized function within the core. This large grid-like structure necessitates a very high mesh density to accurately capture the electromagnetic behavior in the THz band. Consequently, this increased mesh density significantly raises the demand for computational resources. To address this, in HFSS simulation, we represent them by condensing them into a single sheet of copper.

## 6. Simulation and Results

### 6.1. On-Chip Antenna Simulation inside a Chip Package

In the simulation, all the cores (A, B, C, D, E, F, G, H, I) have the same configuration of a zigzag and dipole antenna. However, only cores A, B, C, E, and I were selected because they represent the expected worst-care scenario.

The zigzag and dipole antenna is optimized to have a reflection coefficient ($S_{11}$) of less than 10 dB at 462.5 GHz and 446 GHz, respectively, as illustrated in Figure 5. All the transmissions at the 462.5 GHz (for zigzag antenna) and 446 GHz (for dipole antenna) frequency bands in this paper are measured with respect to Antenna A.

From Figure 7, we can see that the transmission coefficients in the case of a zigzag antenna at 462.5 GHz, the $|S_{21}|$ between Antenna A and Antenna I, and that between Antenna A and Antenna C are −33 dB and −41 dB, respectively, which highlight that the transmission gets reflected more from the enclosed surface of the chip in the case of Antenna I.

A similar case can be observed from Figure 8, from which we can see that the transmission coefficient in the case of a dipole antenna at 446 GHz, the $|S_{21}|$ between Antenna A and Antenna I, and that between Antenna A and Antenna C are −28.26 dB and −28.98 dB, respectively, which are quite similar. Despite Antenna I being farther away, this indicates that the waves get more reflected from the structure of the chip in the case of Antenna I.

Furthermore, this observation is verified by the electric field distribution across the surface of the chip as shown in Figure 9 and Figure 10 for zigzag and dipole antennas, respectively. The cross-sectional view for both antennas shows that there is a high density of e-field plots on the surface of the chip where the antenna is excited and the low-intensity waves become more important to other cores. In this case of intra-chip communication, the mode of propagation is the surface waves as confirmed in [48]. As can be seen from Figure 9 and Figure 10, the E-field distribution of the dipole antennas is more distorted than that of the zigzag antenna. Therefore, when designing on-chip antennas for the intra-chip communication among the cores in a multi-core processor, it is important to consider the analysis of the field distribution. In addition, if the directional or phased array antenna were used for the communication, there might be diffraction causing beams to distort due to the small components inside a chip that ultimately reduces the gain.

Here, the field distribution differs at different frequencies because each antenna resonates at a distinct frequency with a maximum reflection coefficient.

### 6.2. Transmission Gain in Intra-Chip Communication Channel

There is no relative motion between the cores in a multi-core chip; thus, we regard the channel being examined as time-invariant. Transmission gain refers to the cumulative decibel value of the transmit and receive antenna gains combined with the path gain. When measured, it essentially represents the scattering parameter $|S_{21}|$, which quantifies the gain from port 1 to port 2.

Using the Friis transmission equation [49], one can calculate the transmission gain.
(1)$$G_{a}=\frac{\left|S_{21}\right|}{(1-|S_{11}{|}^{2})(1-|S_{22}{|}^{2})}=G_{r}G_{t}{\left(\frac{\lambda }{4\pi r}\right)}^{2}e^{-2\alpha r}=\frac{P_{r}}{P_{t}}$$
where $|S_{21}|$ is the transmitter and receiver coupling, $|S_{11}|$ and $|S_{21}|$ correspond to the reflection coefficient at both ends, $G_{r}$ and $G_{t}$ denote the receiver and transmitter gain, $P_{r}$ and $P_{t}$ represent the received and transmitted power, λ is the operating wavelength (648 µm and 672 µm in the case of zigzag and dipole antenna, respectively), α is the attenuation factor, and *r* is the separation distance between the transmitter and receiver.

These antennas operate within a fixed environmental structure, which introduces limitations for measurement. Hence, the channel model is closely related to the antenna design and the physical structure of the chip. Thus, for the intra-chip channel modeling, we perform a series of measurements for transmission gain (G_*a*_) between antennas as a function of distance. All measurements are conducted at the resonant frequency of both antennas. The data are then post-processed using Equation (Equation 1), and the results are shown in Figure 11.

In this case, we used two different die materials, i.e., polyimide and silicon dioxide, to observe the gain between the antennas. Due to the high resistivity of the silicon, we can observe the less gain between the antennas from Figure 11. The measurement is restricted to a distance of around 19 mm because the maximum separation of a 13 × 13 mm^2^ chip comes to 18.38 mm if the antennas are placed central to each core.

The probability density function (PDF) based on the variation of $G_{a}$ is shown in Figure 12. It is noteworthy that the path loss varies from around −40 dB to +35 dB for the simulated THz band. A dipole antenna under silicon dioxide has more variation. This study is helpful in determining link budget and channel gain estimation for a particular configuration based on antenna locations and positions.

### 6.3. Link Budget Estimation for Intra-Chip Communication

Figure 7 and Figure 8 illustrate $S_{21}$ versus frequency for the antennas under investigation, with Antenna A as the reference. In this context, ${\mathrm{IL}}_{BA}$, ${\mathrm{IL}}_{CA}$, ${\mathrm{IL}}_{EA}$, and ${\mathrm{IL}}_{IA}$ represent the insertion loss (IL) with respect to Antenna A. To define the bandwidth, we consider the range of frequencies where the changes in insertion loss ${\mathrm{IL}}_{iA}<2$ dB (where i = B, C, E, I) are within this limit. The 2 dB threshold indicates an approximate value for a channel without significant distortion [50].

The maximum channel bandwidth in the case of a dipole antenna are as follows: Antenna B is 9 GHz (379–388 GHz), Antenna C is 7 GHz (499–506 GHz), Antenna E is 8 GHz (530–538 GHz), and Antenna I is 7 GHz (376–383). The insertion for the dipole antenna in the 2 dB threshold is shown in Figure 13. Similarly, the maximum channel bandwidth in the case of a zigzag antenna are as follows: Antenna B is 2 GHz (468.5–470.5 GHz), Antenna C is 3 GHz (478.5–481.5 GHz), Antenna E is 4 GHz (472–476 GHz), and Antenna G is 2 GHz (474–476 GHz). The insertion loss for the zigzag antenna having values under the 2 dB threshold is shown in Figure 14. All the frequencies fall under the −10 dB reflection coefficient values of the respective antenna.

Based on Figure 13 and Figure 14, we established a dedicated channel for each pair of antennas. In the context of analyzing the binary OOK modulation, we may infer that SNR_*min*_ = 24 dB is the minimum SNR required for an error probability of $10^{-14}$. This error probability measures how likely it is that noise, interference, or other communication channel imperfections could cause the received signal to be misinterpreted. Subsequently, we calculate the necessary bit energies for transmission and reception as follows [50].

Noise power is given by the following:(2)$$P_{n}=-174\;\mathrm{dBm}/\mathrm{Hz}+10\log_{10}\left(BW\right)+NF$$
where −174 dBm/Hz is the thermal noise floor at room temperature, with BW as the bandwidth (in Hz) and NF as the noise figure (10 dB). The received power ($P_{rx}$) (dBm) and transmitted power ($P_{tx}$) (dBm) are given by the following:(3)$$P_{rx}=P_{n}+SNR_{\min}\;\mathrm{and}\;P_{\mathrm{tx}}=P_{rx}+\mathrm{IL}$$

Finally, the energy per bit for the receiver and transmitter are calculated by the following:(4)$$E_{rx}=\frac{P_{r}}{R_{b}}\;\mathrm{and}\;E_{tx}=\frac{P_{t}}{R_{b}}$$
where $P_{r}$ and $P_{t}$ are in watt.

With the corresponding insertion loss and bandwidth of each antenna with Antenna A as a reference, using (Equation 2)–(Equation 4), the energy per bit for a transmitter and receiver is computed along with the bit rate, as shown in Table 2.

In Table 2, when Antenna A is a dipole, Antennas B, C, E, and G are also dipole antennas. Similarly, when Antenna A is a zigzag, Antennas B, C, E, and G are zigzag antennas as well. According to our 2 dB threshold assumption, based on the simulated results, the dipole antenna demonstrates a wider bandwidth compared with the zigzag antenna, resulting in a higher bit rate for the dipole antenna. Typically, the target energy expenditure required for WNoC communication to remain competitive with wired communication is less than 1 pJ/bit. This energy includes the energy consumed by all transceiver devices. The data obtained for our best-case channel, as shown in Table 2, indicate that even for the maximum IL of −35.08 dB for the dipole antenna and −30.84 for the zigzag antenna, the required energies are $4.04\times 10^{-15}$ J and $1.52\times 10^{-15}$ J, respectively. These values fall below the targeted energy expenditure threshold.

Since Ansys HFSS only provides the electromagnetic wave propagation behavior in our design simulation, proper testing of the signal integrity of the WNoC transceiver requires hardware manufacturing. Some of the tests and measurements we can analyze include impedance matching, noise interference testing, channel characterization using a Vector Network Analyzer (VNA) or a Time Domain Analyzer (TDA), cross talk, reflections, conducting tests in various substrate environment conditions, and RF performance (transmitter power, responsivity, noise equivalent power (NEP)) [32].

### 6.4. Inter-Chip Communication

In this section, we present the transmission characteristics of the inter-chip communication by using the antennas discussed in this paper. The architecture for the inter-chip communication is shown in Figure 15. Here, we use the same antenna mentioned in the paper earlier to characterize the transmission in the inter-chip communication scenarios with the flip-chip packaging model. The distance between the chips is 14 mm by using FR408 ($\epsilon_{r}$ = 3.66 and tanδ = 0.012) PCB. The bottom of the PCB is a perfect electric conductor (PEC).

The diameters of the solder balls in the grid are 0.1mm, and their pitch is 0.25 mm, respectively, with a total of 25 solder balls. The $|S_{11}|$ for both antennas is shown in Figure 16. Here, the slight change in inter-chip $|S_{11}|$ compared with intra-chip $|S_{11}|$ is due to the surrounding structure of the inter-chip configuration. In order to reduce the computational complexity, only the antennas on the edges are computed. The transmission between Antenna A and Antenna B is shown in Figure 17. It is evident that the transmission coefficient is approximately −60 dB for the zigzag antenna, while it measures −43 dB for the dipole antenna. These higher losses could pose challenges in practical applications. However, we have provided a calculation detailing the feasibility of communication with high insertion loss in Appendix A.

Furthermore, Figure 18 and Figure 19 show the E- and H-plane of the dipole and zigzag antenna in the case of inter-chip communication. The dipole antenna’s 2D radiation pattern shows that it radiates electromagnetic energy more efficiently into two opposite directions perpendicular to the antenna’s axis, but the zigzag antenna’s 2D radiation pattern is more distorted and non-symmetrical with respect to the antenna’s axis.

We investigate further to observe the variation in electric field distribution across the inter-chip structure. The top view and side view for both dipole and zigzag antennas are shown in Figure 20 and Figure 21, where only Antenna A is excited. From the side view, we can see that the radiated field is reflected back from the heat sink (used aluminum for simulation) and bounces back from the PCB structure. Under the same identical structure, the maximum electrical field magnitude is greater for the zigzag antenna than for the dipole antenna. The geometry of the inter-chip communication model, the die substrate, and the operating frequency all affect the electric field distribution.

In addition, Figure 22 shows the simulated magnitude of an electric field along the line connecting Antenna A and Antenna B. We can observe the rise of an electric field at around 4.5 mm (for the zigzag antenna) and 6 mm (for the dipole antenna), which is due to the reflected electric field from the bottom PCB structure.

## 7. Conclusions

This paper presents the transmission characteristics of on-chip THz antennas for intra- and inter-chip communication. We have investigated two simple antennas, namely, dipole and zigzag antennas, both resonating in the THz band, and studied them in the presence of densely packed multi-layer flip-chip package environments. Through simulation in Ansys HFSS, we have characterized the transmission, studied the electric field distribution inside a chip and path loss, and provided some basic link budget examples using simulated values. The simulated path loss highlights the improvements of using a polyimide substrate compared with silicon dioxide, and the dipole antenna exhibits more path loss compared with the zigzag antenna. However, according to the bandwidth threshold defined in our paper, a dipole antenna has more bandwidth than a zigzag antenna for intra-chip communications. Similarly, based on the simulations for intra-chip communication, the zigzag antenna demonstrates a more distributed form of electric field radiation reaching up to the receiving antenna compared with the dipole antenna. Overall, the choice of an antenna depends upon the materials used for the substrate, the architecture of the chip, and its placement.

Furthermore, it is necessary to investigate channel estimation, modulation, demodulation, and bit error rate calculation for full-wave communication within a flip-chip package. This emphasizes the application of compact on-chip THz transceivers integrated into chip packages, making it a viable technology for wireless interconnections for multi-core CPUs in HPC applications.

## Figures and Tables

**Figure 1 sensors-24-03220-f001:**
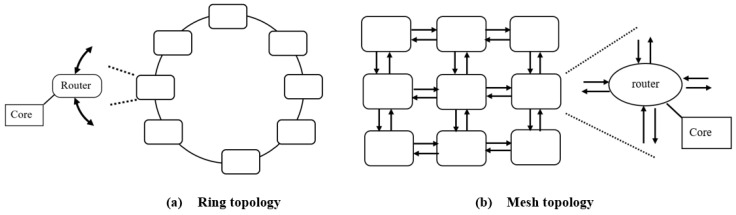
Ring and mesh topology.

**Figure 2 sensors-24-03220-f002:**
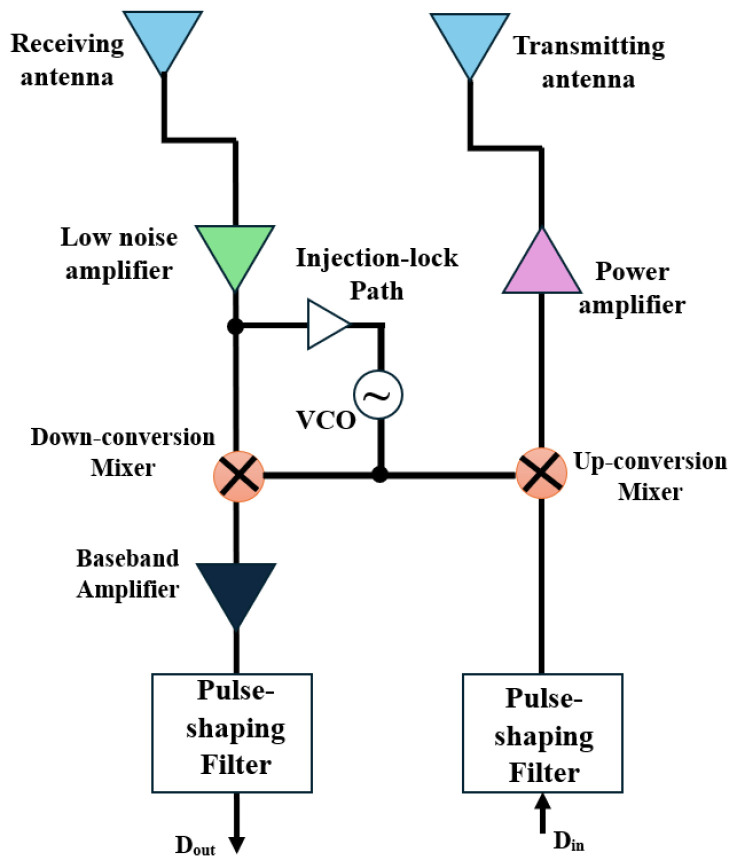
A simple block diagram of OOK transceiver.

**Figure 3 sensors-24-03220-f003:**
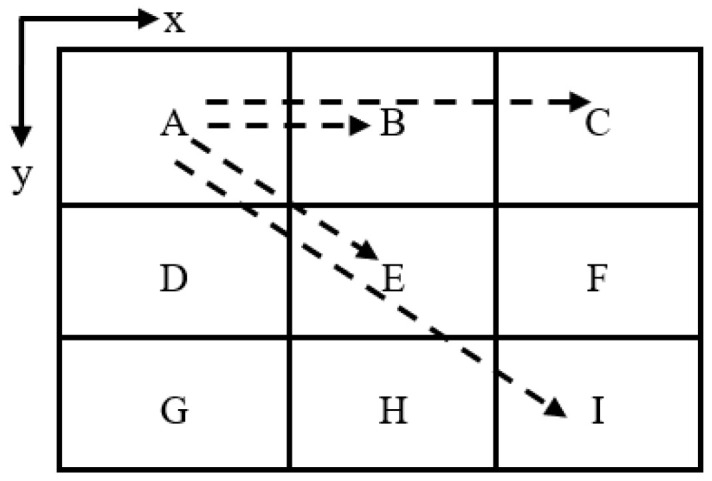
Transmission scenario across cores in a chip.

**Figure 4 sensors-24-03220-f004:**
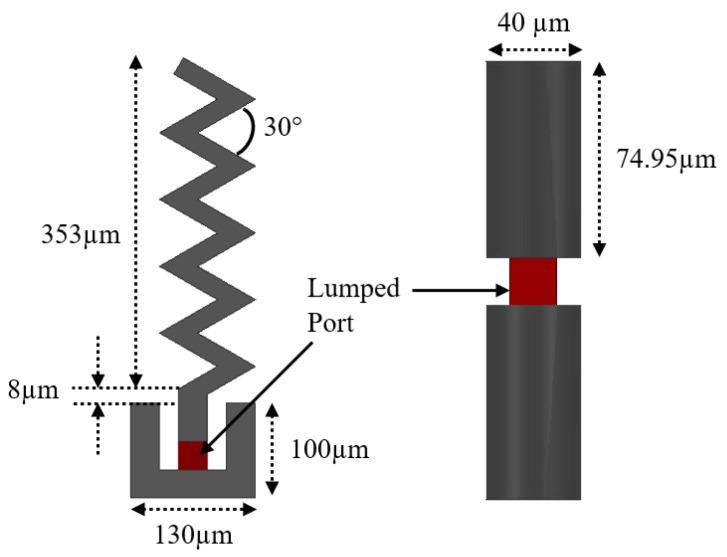
On-chip zigzag and dipole antenna.

**Figure 5 sensors-24-03220-f005:**
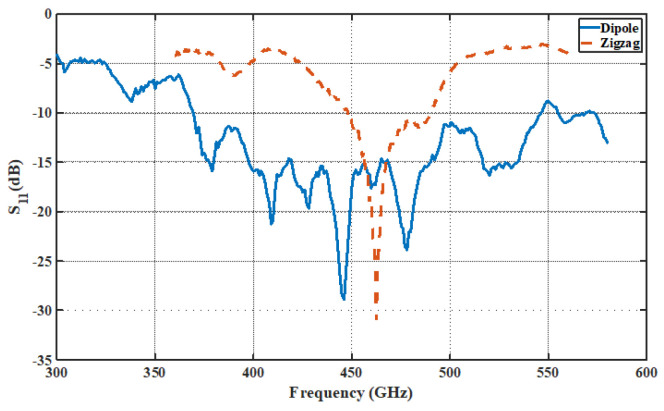
$|S_{11}|$ of dipole and zigzag antenna under polyimide substrate.

**Figure 6 sensors-24-03220-f006:**
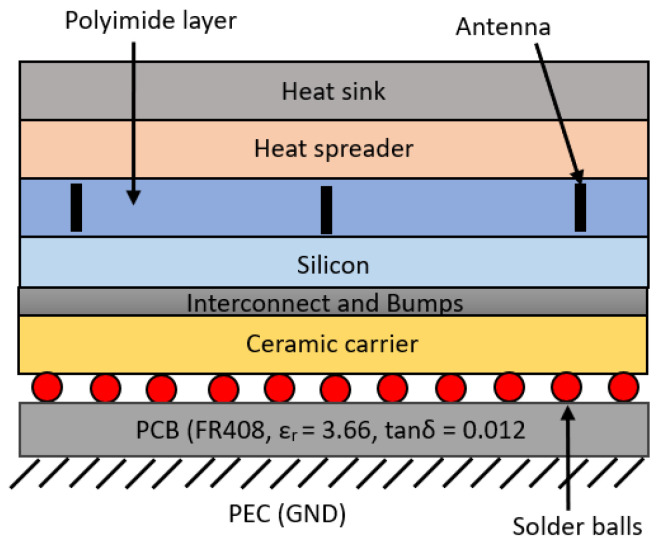
Basic layers inside a multi-chip processor with the proposed antenna placement.

**Figure 7 sensors-24-03220-f007:**
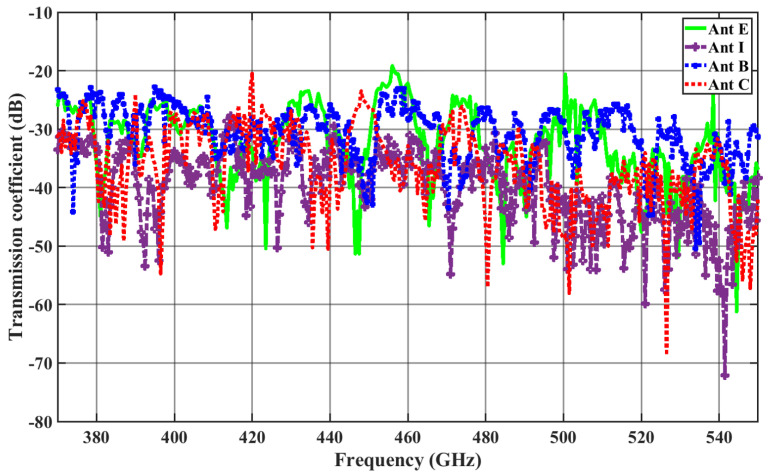
Transmission coefficient for zigzag antennas with Antenna A as the transmitter.

**Figure 8 sensors-24-03220-f008:**
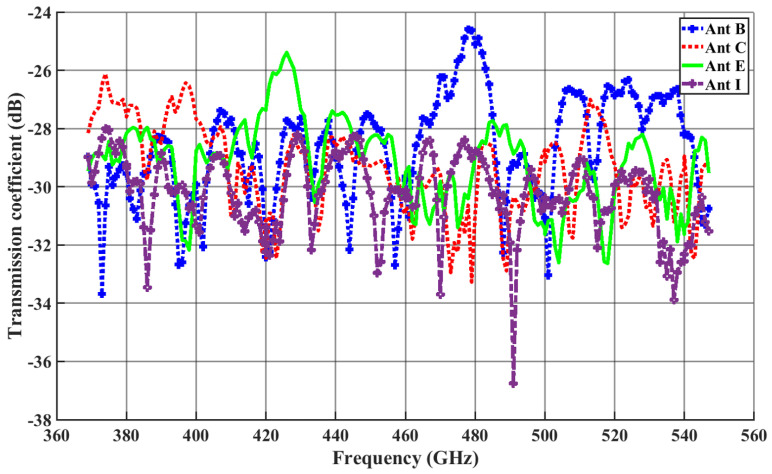
Transmission coefficient for dipole antennas with Antenna A as the transmitter.

**Figure 9 sensors-24-03220-f009:**
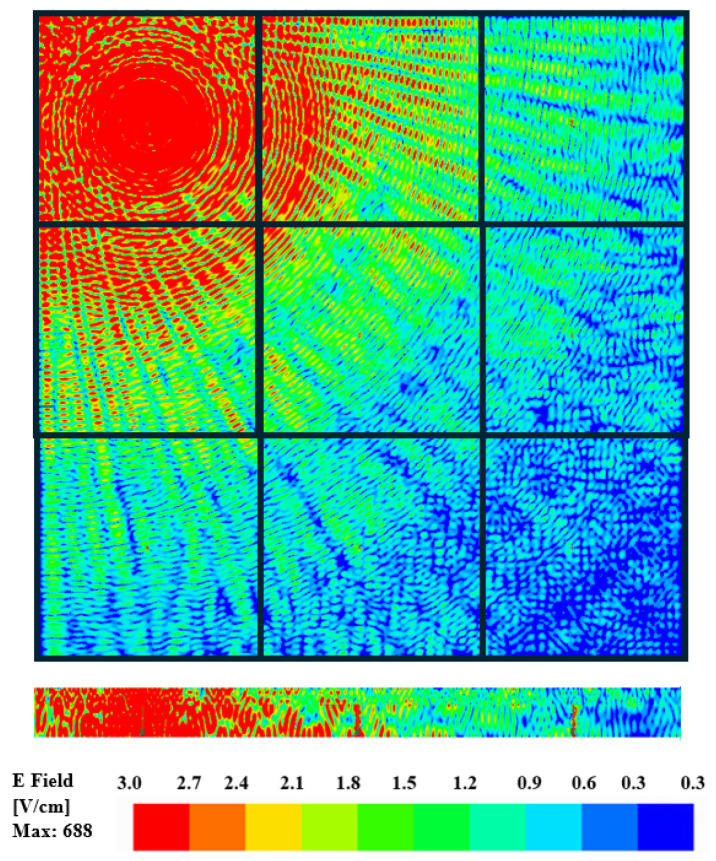
Distribution of electric field magnitude (V/cm) on the die surface when the zigzag antenna is stimulated at a frequency of 462.5 GHz.

**Figure 10 sensors-24-03220-f010:**
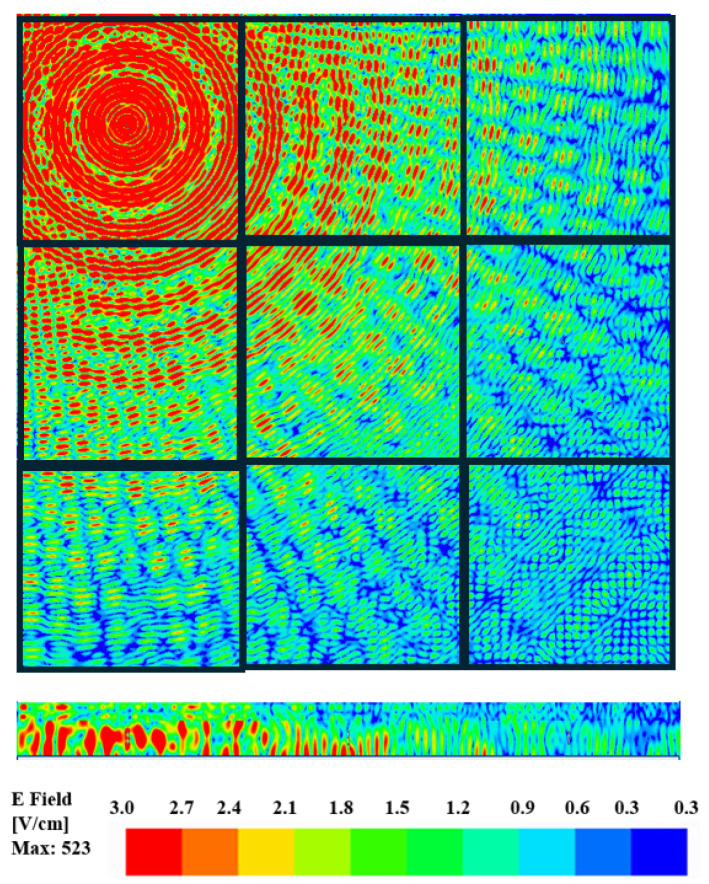
Distribution of electric field magnitude (V/cm) on the die surface when the dipole antenna is stimulated at a frequency of 446 GHz.

**Figure 11 sensors-24-03220-f011:**
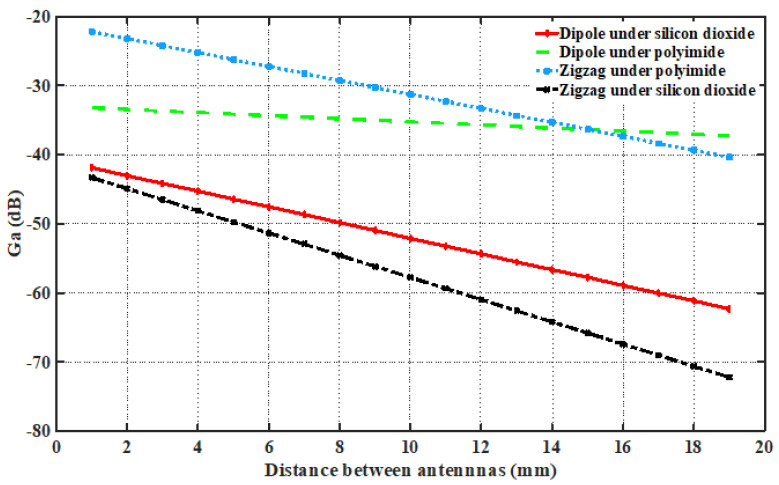
Measured transmission gain (G_*a*_) with Antenna A as a reference.

**Figure 12 sensors-24-03220-f012:**
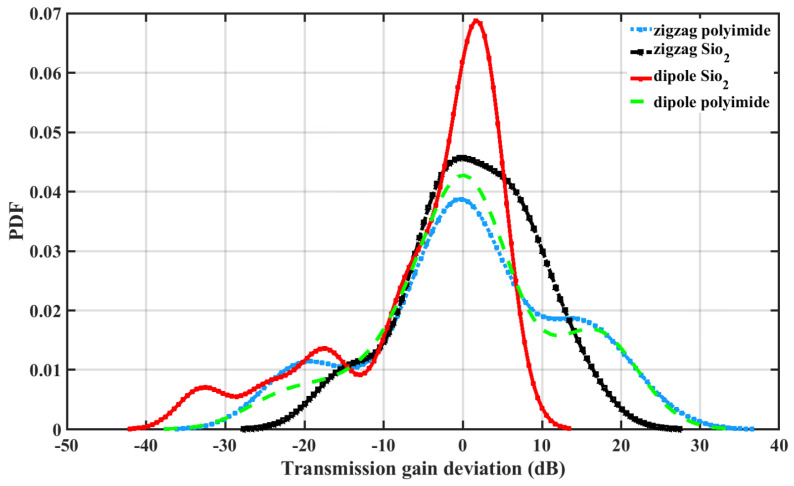
Probability density function (PDF) of transmission gain (G_*a*_) deviation.

**Figure 13 sensors-24-03220-f013:**
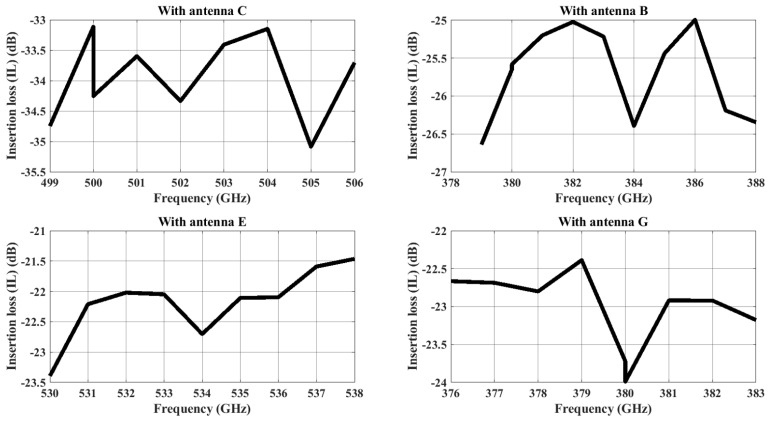
The insertion loss of dipole antenna pairs of less than <2 dB when Antenna A is transmitting.

**Figure 14 sensors-24-03220-f014:**
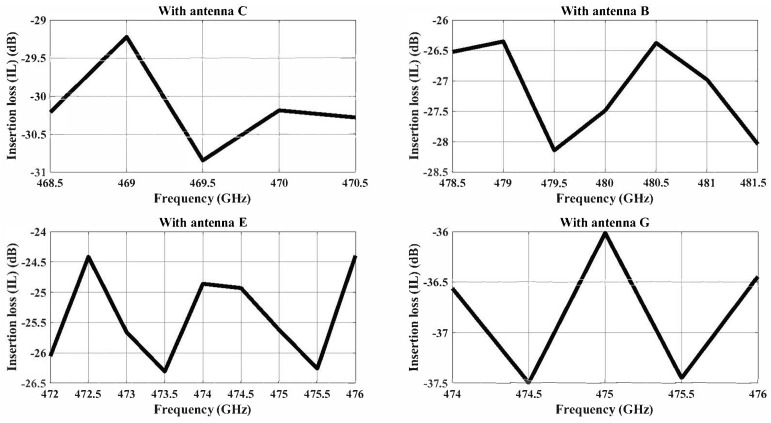
The insertion loss of zigzag antenna pairs of less than <2 dB when Antenna A is transmitting.

**Figure 15 sensors-24-03220-f015:**
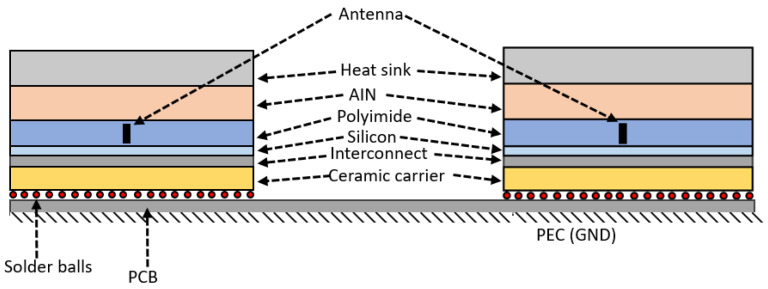
Inter-chip communication configuration.

**Figure 16 sensors-24-03220-f016:**
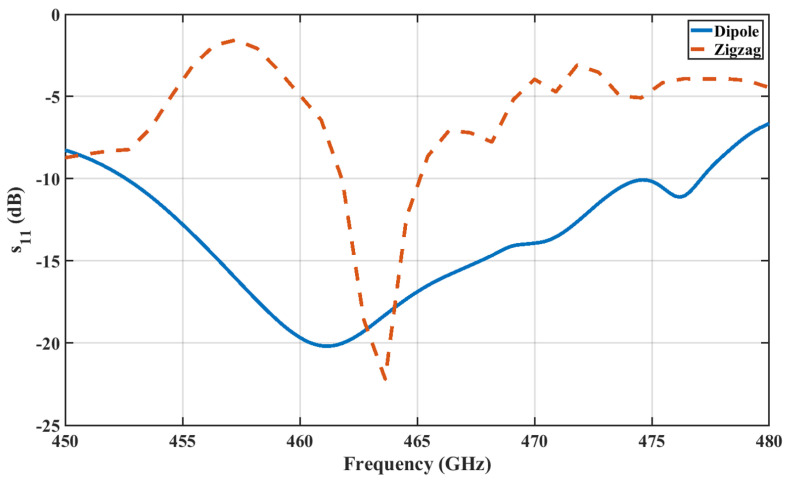
$|S_{11}|$ under inter-chip communication.

**Figure 17 sensors-24-03220-f017:**
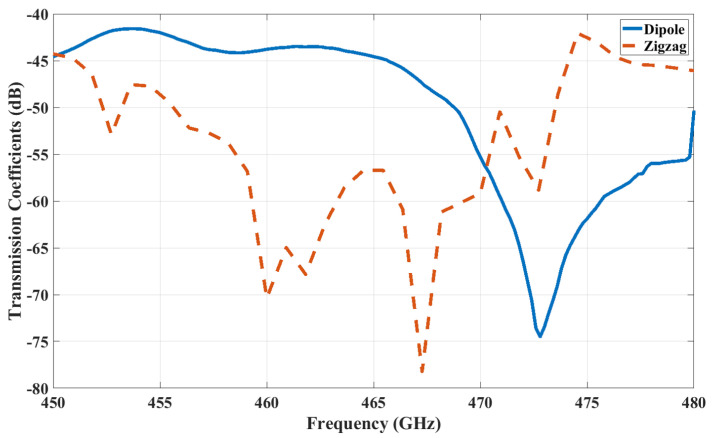
Transmission coefficients ($|S_{21}|$) under inter-chip communication.

**Figure 18 sensors-24-03220-f018:**
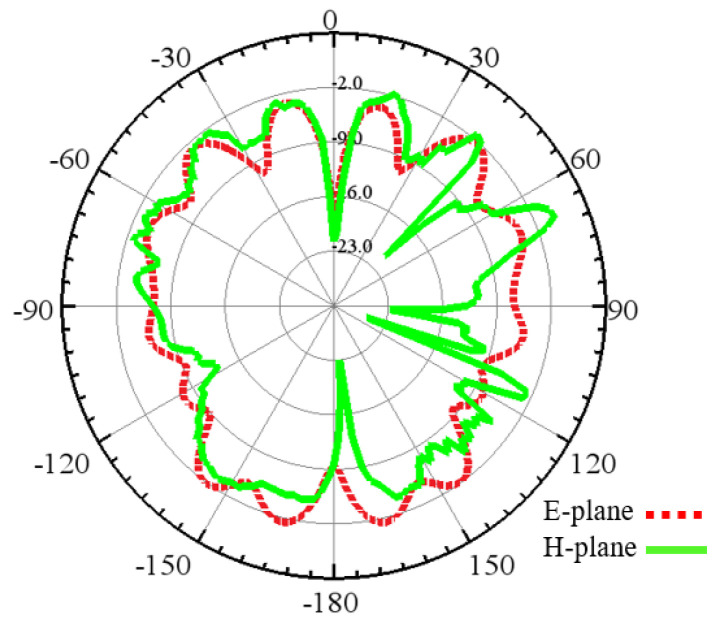
Radiation pattern of the dipole antenna in the E- and H-plane.

**Figure 19 sensors-24-03220-f019:**
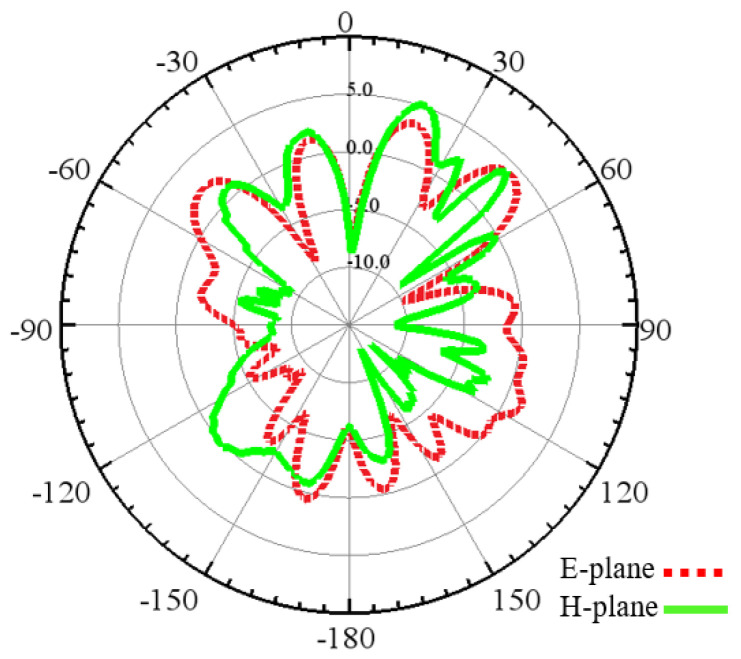
Radiation pattern of the zigzag antenna in the E- and H-plane.

**Figure 20 sensors-24-03220-f020:**
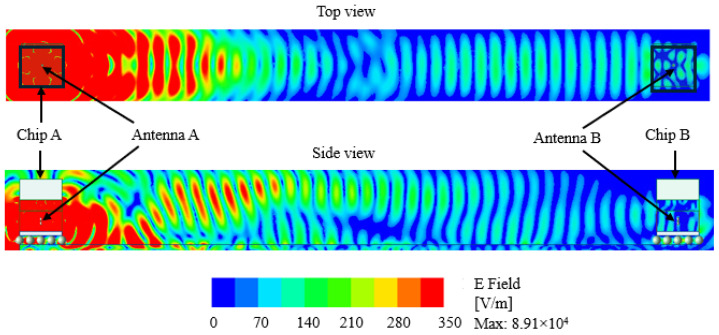
Electric field distribution across the inter-chip communication in the case of the dipole antenna.

**Figure 21 sensors-24-03220-f021:**
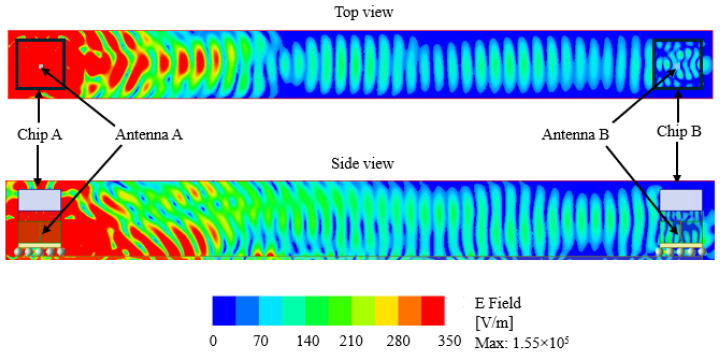
Electric field distribution across the inter-chip communication in the case of the zigzag antenna.

**Figure 22 sensors-24-03220-f022:**
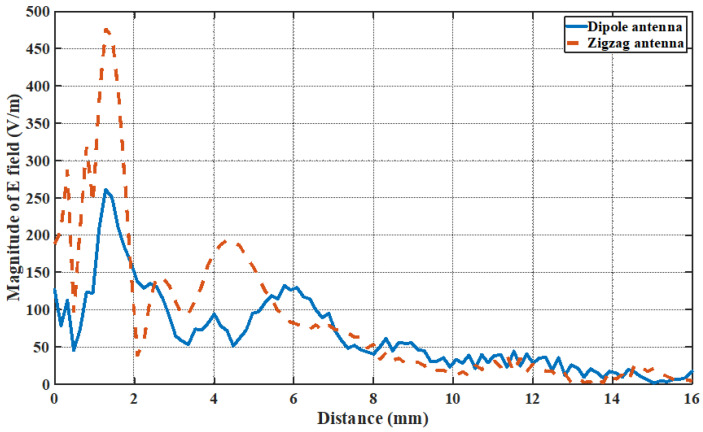
Electric field strength (V/m) along the line that separates Antennas A and B (only Antenna A is switched on).

**Table 1 sensors-24-03220-t001:** Materials and their properties of layers inside a chip package.

Name	Thickness (mm)	Material/Properties
Heat sink	0.5	Aluminum
Heat spreader	0.25	Aluminum nitride (AIN) ($\epsilon_{r}=8.6$, tanδ=0.0003)
Polyimide	0.5	$\epsilon_{r}=3.5$, tanδ=0.008
Silicon	0.5	$\epsilon_{r}=11.9$, tanδ=0.2
Interconnect	0.013	Copper
Ceramic carrier	0.5	Alumina ($\epsilon_{r}=9.4$, tanδ=0.006)
Solder balls	0.32	Perfectly electric conductor
Dipole and zigzag antenna	-	Copper

**Table 2 sensors-24-03220-t002:** Link budget with Antenna A as the transmitter with two scenarios: (a) using dipole antenna and (b) using zigzag antenna.

	Dipole					Zigzag				
Receiving	BW	IL	Bit Rate	Energy per Bit (J/bit)		BW	IL	Bit Rate	Energy per Bit (J/bit)	
Antennas	(GHz)	(dB)	(Gbps)	Tx ($E_{tx}$)	Rx ($E_{rx}$)	(GHz)	(dB)	(Gbps)	Tx ($E_{tx}$)	Rx ($E_{rx}$)
B	9	−26.64	71.81	5.78$\;\times \;10^{-16}$	1.25$\;\times \;10^{-18}$	2	−28.14	15.96	8.17$\;\times \;10^{-16}$	1.25$\;\times \;10^{-18}$
C	7	−35.08	55.85	4.04$\;\times \;10^{-15}$	1.25$\;\times \;10^{-18}$	3	−30.84	23.94	1.52$\;\times \;10^{-15}$	1.25$\;\times \;10^{-18}$
E	8	−23.39	63.83	2.74$\;\times \;10^{-16}$	1.25$\;\times \;10^{-18}$	4	−26.3	31.91	5.35$\;\times \;10^{-16}$	1.25$\;\times \;10^{-18}$
G	7	−23.99	55.85	3.14$\;\times \;10^{-16}$	1.25$\;\times \;10^{-18}$	2	−37.49	15.96	7.03$\;\times \;10^{-15}$	1.25$\;\times \;10^{-18}$

## Data Availability

Data are contained within the article.

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
