# Peer review of "Design and Modeling of a Terahertz Transceiver for Intra- and Inter-Chip Communications in Wireless Network-on-Chip Architectures"

_sensors, 2024, doi:10.3390/s24103220_

Round 1

Reviewer 1 Report

Comments and Suggestions for Authors

I have reviewed the manuscript titled “Inter- and Intra-Chip Wireless Communications using On-Chip Terahertz Antennas in a Multi-core Processor” prepared by Paudel et al. In this work, the authors investigate the capabilities of the THz waves for ultra-short-distance communication within and among the chips in simulation. The problems with the manuscript are as follows:

1. As shown in Fig. 17, the transmission coefficients under inter-chip communication is lower than -40 dB, which is too high for practical application.

2. There is another challenge for practical application. How to generate THz waves using this antenna?

Comments on the Quality of English Language

It is good.

Reviewer 2 Report

Comments and Suggestions for Authors

The manuscript entitled "Inter- and Intra-Chip Wireless Communications using On-Chip Terahertz Antennas in a Multi-core Processor" has paid much attentions on the wireless communication performance of the physical transmission for dipole and zigzag antennas of different layouts for intra- or inter-chip packages. Many results are given and there are some comments for references:

1. The full-wave simulation method used is ANSYS HFSS, so how could you please further consider the Signal Integrity for the Tx/Rx communication in reality?

2. In recent years, the interconnections between Chip/Chiplets seems much more interesting. What do you think of the method used in the manuscript for this area analysis?

3. In Page 5 of 16, the ending two sentences of the first paragraph from line 156 to 161 seem repeating and can be shortened.

4. The model as shown in Fig. 6 seem much more simpler than that (Fig. 1) of Ref.[19], which is more practical and closed to the real flip-chip package.

5. As for the Section 4, there seem no show fabrication and real measurement, all the results given are from the simulation ones. Right?

6. In Page 7 of 16, for Line from 197-198, as depicted in Fig. 8, the |S21| between Ant A and Ant I, and that between Ant A and Ant C are -28 dB and -29 dB, which are different from the manuscript. To see whether the data is right or not.

7. As for the Fig. 11, please judge whether the data are measured ones or simulated ones?

8. As for the analysis sequence for the intra-chip at first and the inter-chip thereafter, the title of manuscript of "Intra- and Inter-Chip Wireless Communications Using on-Chip Terahertz Antennas in a Multi-Core Processor" is more preferable.

Comments on the Quality of English Language

Well written but more accurate data should be presented.

Reviewer 3 Report

Comments and Suggestions for Authors

Overall, I was very interested in the article. I would like it to be published. But there are a number of criticisms to it:

1. The statement "In the wired connections of multi-core CPUs there are two types of topologies, i.e. 2D mesh and ring topologies." is rather strange.

First, the authors should cite ring topologies like [https://doi.org/10.1016/j.procs.2015.04.190] and their developments like ring circulant [https://doi.org/10.1016/j.heliyon.2019.e01516] or spidergon [https://doi.org/10.1007/978-1-4614-4274-5_7, https://doi.org/10.1109/DATE.2006.243841], etc.

Second, these are not the only two topologies. There are actually dozens of them, and should have been mentioned.

For example, Fig. 1 (b) is very unfortunate in this context. It does not look like a Flattened butterfly as it is commonly figured, and even in its flat form, the figure is poorly done, failing to demonstrate the principle of long links. Note Figure 4 in [https://dx.doi.org/10.14569/IJACSA.2016.071010], which illustrates the Flattened butterfly much better.

And as this figure is now, it looks more like a torus topology, where instead of individual nodes, there are clusters of nodes.

2. When you write about "long-range links inserted into the existing mesh" the Diametrical Mesh topology [https://doi.org/10.1007/978-3-642-31600-5_65] comes to mind, I was surprised you didn't mention it.

3. Add appropriate references when you talk about "photonics and optical interconnects".

4. I was expecting the Related Works section to be a short overview of existing Wireless NoCs solutions. Add a paragraph about that.

5. What is the difference between WNoC and WiNoC? Both forms are encountered.

6. Add a paragraph about Contribution to the Introduction.

7. Starting from section 3.2 I was disappointed. Because there went a part as if written by another person and about things not related to the first part of the paper. Analysis and modeling of electric fields and types of antennas can certainly be interesting, but for specialists in another field. To me, as a specialist in high-level development of networks-on-chip, – no.

At the same time, Abstract and the title of the article do not say at all that the article is about magnetic fields and antenna modeling.

What is needed is an additional section (discussion) where you tie the two different parts of the article together.

Is there any way to estimate the delay of flit transmission over a wireless channel in the hops of a wired channel? Maybe it doesn't make sense to create a WNoC at all if the latency and throughput of the wireless channel will be incommensurable with the wired channel?

How to take into account the peculiarities of WNoC at the level of network topology? What routing algorithms should be there? How secure is such a communication channel? Is there a need for noise-resistant coding? And information security? Can a WNoC be disrupted by external influence (e.g., magnetic field)? And so on.

8. The conclusions do not need to describe what you did. There should be specific results. I read the conclusions several times, but I never realized which type of antenna is the best suited.

By the way, I may not have read this part very carefully, but was there a justification for the choice of antenna types and an analysis of analogs made before you?

Round 2

Reviewer 1 Report

Comments and Suggestions for Authors

The authors did not answer my question. My question is “How to generate THz waves using this antenna”, but not “How to generate THz waves”. Section 3.1 only answer “How to generate THz waves”, but not my question.

If the authors use another kind of THz resources to couple THz signals into the antenna, there are some problems. First, the insertion loss is very high. Second, it is quite difficult to couple the signal only to the target location, but not to the other locations.

The authors claimed “the result of transmission power of -3.5 dBm is quite feasible for modern communication”. However, they only compared the intensity of THz signals with that of noise. Then, there is another problem. How to detect the THz signals with intensities slightly higher than that of noise using such a structure?

Comments on the Quality of English Language

The Quality of English Language is ok.

Reviewer 2 Report

Comments and Suggestions for Authors

There are still some suggestion for further revision. FYI.:

1. In the Abstract, there is lack of the certain conclusion for the performance parameter values about design and modeling of the proposed THz Tx/Rx in detail.

2. Do you think the Signal Integrity analysis for the Tx/Rx communication in reality is hard to be achieved only by using the full-wave simulation Frequency-Domain Differential-Equation method solver ANSYS HFSS? I see the supplementary in Section 6.3 on page 12, but there is no more analysis based-on data.

3. There seems no feedback about the interconnections between Chiplets which is becoming more and more popular in current academic area.

4. As for the Section 4, there seem no show fabrication and real measurement, all the results given are from the simulation ones. Could you please do at least one simplest verification for further check?

Comments on the Quality of English Language

Good enough.

Reviewer 3 Report

Comments and Suggestions for Authors

The authors took my comments into account and revised their article quite well. I think it can be published. 

I do not like only one point, which I recommend the authors to correct in the final version without additional review.

Multi-core Processor -- this is a processor with multiple processing cores. It is not necessarily a NoC. Some ancient core-two-duo, that's a Multi-core Processor too.

In my opinion, in the context of the article we should use the term MPSoC -- Multiprocessor System-on-Chip or stay within the term WNoC everywhere.
